# Enrofloxacin Alters Fecal Microbiota and Resistome Irrespective of Its Dose in Calves

**DOI:** 10.3390/microorganisms9102162

**Published:** 2021-10-17

**Authors:** Ashenafi Feyisa Beyi, Debora Brito-Goulart, Tyler Hawbecker, Brandon Ruddell, Alan Hassall, Renee Dewell, Grant Dewell, Orhan Sahin, Qijing Zhang, Paul J. Plummer

**Affiliations:** 1Department of Veterinary Microbiology and Preventative Medicine, College of Veterinary Medicine, Iowa State University, Ames, IA 50011, USA; afbeyi@iastate.edu (A.F.B.); dgoulart@iastate.edu (D.B.-G.); bruddell@iastate.edu (B.R.); zhang123@iastate.edu (Q.Z.); 2College of Veterinary Medicine, Iowa State University, Ames, IA 50011, USA; tjh3@iastate.edu; 3Department of Veterinary Diagnostic and Production Animal Production, College of Veterinary Medicine, Iowa State University, Ames, IA 50011, USA; ahassall@iastate.edu (A.H.); gdewell@iastate.edu (G.D.); osahin@iastate.edu (O.S.); 4Center for Food Security/Public Health, College of Veterinary Medicine, Iowa State University, Ames, IA 50011, USA; rdewell@iastate.edu; 5National Institute of Antimicrobial Resistance Research and Education, Iowa State University, Ames, IA 50010, USA

**Keywords:** antibiotic resistance, bovine respiratory disease complex, calves, fluoroquinolone/enrofloxacin, gut microbiota, resistome profiles

## Abstract

Enrofloxacin is a fluoroquinolone drug used to prevent and control bovine respiratory disease (BRD) complex in multiple or single doses, ranging from 7.5 to 12.5 mg/kg body weight. Here, we examined the effects of high and low doses of a single subcutaneously injected enrofloxacin on gut microbiota and resistome in calves. Thirty-five calves sourced for this study were divided into five groups: control (*n* = 7), two low dose groups (*n* = 14, 7.5 mg/kg), and two high dose groups (*n* = 14, 12.5 mg/kg). One group in the low and high dose groups was challenged with *Mannheimia haemolytica* to induce BRD. Both alpha and beta diversities were significantly different between pre- and post-treatment microbial communities (*q* < 0.05). The high dose caused a shift in a larger number of genera than the low dose. Using metagenomic ProxiMeta Hi-C, 32 unique antimicrobial resistance genes (ARGs) conferring resistance to six antibiotic classes were detected with their reservoirs, and the high dose favored clonal expansion of ARG-carrying bacterial hosts. In conclusion, enrofloxacin treatment can alter fecal microbiota and resistome irrespective of its dose. Hi-C sequencing provides significant benefits for unlocking new insights into the ARG ecology of complex samples; however, limitations in sample size and sequencing depth suggest that further work is required to validate the findings.

## 1. Introduction

Gut microbiota, consisting of bacteria, archaea, viruses, protozoa, and fungi, play vital roles in animals by conferring colonization resistance against pathogens, modulating immunity, and increasing food-conversion efficiency [1,2]. However, these beneficial commensal organisms are vulnerable to external aggressions, such as antibiotic exposure [3,4,5]. Despite their side effects, antibiotics have significantly improved animal health, welfare, and productivity since their discovery in the mid-20th century [6]. Reports from several countries show that vast volumes of antibiotics, as high as 80% of all antimicrobials used, are applied to prevent and control diseases, as well as to promote growth as feed additives in animal husbandry (see a critical review by Ferri et al. [7]). Moreover, the global consumption of antimicrobials in food animals is expected to increase by 67% of the 2010 consumption level by 2030 [8], prompting a call for strict regulations.

Overuse and inappropriate use of antibiotics in animals have significantly contributed to the disruption of the microbial community, which may have potentially harmful consequences on the host. Furthermore, the misuse of antibiotics selects for antibiotic-resistant bacteria and resistance genes, converting the gut microbiota to a reservoir of recalcitrant multi-resistance strains and genes that could endanger public health by spreading to humans through the food chain and environmental contamination [9,10]. Collectively, the collateral effects of antibiotics on gut microbiota coupled with the unprecedented health threat posed by alarmingly rising antimicrobial resistance (AMR) incidences warrant critically evaluating our antibiotic management practices and strengthening antibiotic stewardship [11].

Enrofloxacin is a fluoroquinolone drug used to prevent and control bovine respiratory disease (BRD) complex, the leading cause of morbidity and mortality in U.S. feedlot cattle [12,13,14]. This drug is recommended in single and multiple doses. Even the single dose comes with a wide range of dosages, ranging from 7.5 to 12.5 mg/kg body weight [15]. Interestingly, dosing regimens have been recognized as impacting gut microbiota differently. For instance, chickens treated with various concentrations of enrofloxacin through the oral route (0, 0.1, 4, and 100 mg/kg body weight) exhibited different outcomes after the course of the medication; microbial diversity was significantly more reduced in the higher dosage groups than in the control group [16]. A high dose of drug administration results in a long exposure time to antibiotics that may exert stronger selective pressure on the gut microbiota. Parenterally administered antibiotics to treat diseases diffuse from plasma to the gastrointestinal lumen, influencing the commensal organisms [17,18,19]. A study shows that the concentrations of enrofloxacin and its metabolite, ciprofloxacin, in the ileum and colon were higher than the plasma and interstitial tissue concentrations for several hours following subcutaneous injection [19]. Protein binding of enrofloxacin diminishes its bioavailability and bactericidal effect [17]; however, it has been demonstrated that 54% and 81% of enrofloxacin and ciprofloxacin, respectively, were unbound in the plasma [15]. The penetration rate of enrofloxacin from the plasma to the ileum and colon is also very high [18,19]. Furthermore, Foster et al. [18] reported that enrofloxacin reached intestinal concentrations capable of decreasing gut bacteria following a single dose subcutaneous injection (12.5 mg/kg) to steers. Taken together, changing the therapeutic dosage of enrofloxacin used for the prevention and control of BRD in feedlot cattle might minimize its collateral damage on the gut microbial community and the development of AMR.

A study done by Ferguson et al. [19] has demonstrated the absence of a significant impact difference on gut microbial diversity between multiple and single therapeutic doses of enrofloxacin injected subcutaneously into steers. However, little is known about the relative effects of high and low single doses of enrofloxacin on fecal microbiota and resistome. Hence, the purpose of this study was to evaluate the relative effects of upper (12.5 mg/kg) and lower (7.5 mg/kg) therapeutic doses of enrofloxacin on fecal microbes and resistome in calves. We hypothesized that the lower therapeutic dose would result in a smaller concentration of enrofloxacin and its metabolite in the intestine, thereby altering the microbial diversity to a lesser extent compared to the higher dose due to the shortened exposure time and lessened selective pressure.

## 2. Materials and Methods

### 2.1. Study Design and Sample Collection

The Iowa State University Institutional Animal Care and Use Committee approved the animal procedures (IACUC 8-12-7432-B). All methods and procedures in the animal study were performed in full compliance with the Committee’s guidelines and regulations.

***Calf procurement and classification to study groups***: Thirty-five male beef calves with approximate ages between 12 and 16 weeks and weight ranging from 73 to 136 kg were procured from a farm located in the state of Wisconsin for this study. They were visually examined by veterinarians and confirmed to be free of signs of disease, including lameness, nasal discharges, dyspnea, obtundation, ophthalmic abnormalities, bloat, or diarrhea, upon arrival at Iowa State University (Ames, Iowa). Following the examinations, the calves were tagged with unique identification numbers, weighed, blocked by their weight, randomly assigned into five groups, and housed in one of five rooms maintained at 20–21 °C (68–70 °F) in a biosecurity Level 2 Livestock Infectious Disease Facility (LIDF) at Iowa State University. They were group-housed in five rooms equipped with an independent airflow system (seven calves in each room) at the LIDF for 28 days. The logistics were arranged in a way that minimizes cross-contamination among rooms. The calves were divided into five groups: the control group received oral inoculation of *Campylobacter jejuni*; two groups of low dose cohort administered with *C. jejuni* orally, and a single dose of enrofloxacin (7.5 mg/kg body weight) subcutaneously; and two groups of high dose cohort administered with *C. jejuni* orally, and a single dose of enrofloxacin (12.5 mg/kg body weight) subcutaneously. The low and high dose groups were further categorized into two groups: one group from each dose was challenged with *Mannheimia haemolytica* to induce BRD, while the remaining two groups (one from each dose) served as control groups (Table 1). Calves were fed mixed grass hay and a pre-mixed calf starter (Heartland Co-op, Des Moines, IA) throughout the study period; access to water was *ad libitum*. None of the calves demonstrated serious health problems that required additional antibiotic administration during the study course.

***Study procedures and sample collections***: After four days of acclimatization, all calves were orally inoculated with *C. jejuni* (a cocktail of laboratory strains obtained from Missouri and Colorado). The inoculum was prepared by combining fresh colonies of *C. jejuni* with Mueller–Hinton (MH) broth. For each calf, 60 mL (~4 × 10^9^ CFU/mL) *C. jejuni* suspension in MH broth was administered using a calf esophageal tube. In short, a halter was put on the muzzle to restrain the calf by tying to a fixed object; the esophageal tube was passed through the mouth to the rumen; 60 mL of *C. jejuni* suspension was gavaged through the tube using a large syringe; and finally, the tube was flushed with 50 mL of water. Eight days following *C. jejuni* inoculation, calves in groups designated as sick-low dose and sick-high dose were inoculated with *M. haemolytica* suspended in PBS (20 mL per calf, 5 × 10^8^ CFU/mL) via trans-tracheal injection using a catheter to induce BRD, as described in our publication [20]. The calves were monitored for signs of BRD, such as elevated temperature, eye, and nasal discharges, ear droop or head tilting, cough, and changes in breathing, eating, and ambulatory daily, until signs subsided. Eight days after intratracheal inoculation of *M. haemolytica,* a single dose of enrofloxacin (BAYTRIL™ 100, Bayer Animal Health, Shawnee Mission, KS, USA) was injected subcutaneously into the calves in the low dose groups (7.5 mg/kg) and the high dose groups (12.5 mg/kg) in the neck. Fecal samples were collected directly from the rectum to 50 mL screw-cap tubes from all study calves on days 2, 7, 16, 21, 22, 24, and 28, four times before the treatment (referring to enrofloxacin injection) and three times after the treatment. The fecal samples were stored at −80 °C for laboratory analyses. As per AVMA Guidelines on Euthanasia, the calves were euthanized with a penetrating captive bolt gun on the 28^th^ day of the study time [21].

### 2.2. DNA Extraction and 16S rRNA Gene Sequencing

The 16S ribosomal RNA gene was extracted from 245 fecal samples using ZymoBIOMICS™ kits (Irvine, CA, USA), following the instructions of the manufacturer. Briefly, 200 mg of feces from the fresh fecal sample (collected directly from rectum) kept on ice was transferred to a 2 mL ZR BashingBead™ lysis tube and stored at −80 °C until use. When frozen samples were processed, the samples were first thawed at room temperature for approximately 30 min before being transferred to the lysis tube. Then, 250 µL deionized sterile water, 750 µL lysis solution, and 50 µL proteinase K were added to the lysis tube loaded with 200 mg feces. After that, the samples mixed with the reagents were processed by a bead beater for 10 min followed by incubation in a water bath at 55 °C for at least 30 min. Then, the lysis tubes were centrifuged in a microcentrifuge at 10,000× *g* for 3 min. The supernatant was harvested to columns and then washed with DNA Wash Buffers 1 and 2. The final product was eluted with DNase/RNase-free water. The concentration of eluted DNA was measured first by NanoDrop 3300 Fluorospectrophotometer (Nanodrop Technologies, Wilmington, DE, USA) and confirmed by Qubit fluorometer (Turner BioSystems, Sunnyvale, CA, USA). All the DNA extracts were normalized, transferred to 96-wells plates, and submitted for sequencing to the DNA Facility of Iowa State University. The V4 hypervariable region of the bacterial 16S rRNA gene was amplified using a universal 16S forward primer (515F: GTGYCAGCMGCCGCGGTAA) and a reverse primer (806R: GGACTACHVGGGTWTCTAAT). The sequencing was done following the Earth Microbiome Project protocol using the Illumina MiSeq platform (2 × 250 paired-ends) in a single flow cell lane. Two community standards were included as controls.

### 2.3. Bioinformatics and Statistical Analysis

For the bioinformatic analysis, QIIME 2 was used. The sequence data obtained from the sequencing facility were demultiplexed, and thereafter, denoised using the DADA2 plugin to remove singleton and chimeric reads. The sequencing depth was assessed by plotting alpha rarefaction curves before proceeding to the computation of alpha and beta diversity metrics. Samples with a low number of reads were excluded from the alpha and beta diversity analyses. The richness and evenness of the microbial diversities were evaluated using observed operational taxonomic units (OTUs) and the Shannon index, respectively. Similarly, a Bray–Curtis dissimilarity index was produced, and a principal coordinate analysis was plotted to assess diversities between groups. The taxonomic assignment of representative amplicon sequence variants (ASVs) was conducted with the DADA2 against the Silva ribosomal RNA gene database (release 132).

Alpha diversities were compared between study calf groups and pre- and post-treatment samples by Kruskal–Wallis and Wilcoxon rank-sum tests using R statistical software. Similarly, variations of relative abundances of bacteria between groups were assessed using the same non-parametric statistical tests. An analysis of the composition of microbiomes (ANCOM), demonstrated to overcome the correlation between the compositions of bacteria [22], was conducted to minimize the effects of the compositionality of the relative abundances of bacteria, and to identify bacterial taxa that define groups using QIIME2. The ANCOM analysis was performed at phylum, family, and genus levels. Linear mixed-effects models were used to assess the effects of different covariates on the relative abundances of bacterial taxa. For statistical comparisons, *p* < 0.05 was considered significant.

### 2.4. Metagenomic Hi-C ProxiMeta

To assess the impact of enrofloxacin on resistome profiles of gut microbiota in the calves, metagenomic Hi-C ProxiMeta was employed. This method has been shown to be effective in identifying ARGs along with their reservoirs in high-biomass environmental samples, such as cattle feces [23,24]. Four groups of calves, control, low dose healthy, high dose healthy, and high dose sick, were included in this analysis. An equal number of fecal samples was pooled together from each calf sample (*n* = 7) in a group for the pre-treatment sample (Day 21) and the post-treatment sample (Day 28), separately. The library preparations and bioinformatic analyses were conducted as described in our recent publication [25]. To compare the changes in ARG numbers and host ranges, the cluster numbers were normalized to account for the difference in cluster numbers between pre-and post-treatment samples.

### 2.5. Quantification of Selected Antibiotic Resistance Determinants

Real-time quantitative PCR (qPCR) was run to evaluate the quantitative changes of selected ARGs following enrofloxacin administration. Pooled samples from three calf groups, such as control, low dose, and high dose healthy calves, were used to compare pre-and post-treatment levels of the ARGs. For this assessment, *tet*W, *tet*O, *tet*X, *erm*B, and *erm*F levels were quantified in the samples. The primers used for these ARGs are presented in Table 2, and they were previously used for the same purpose [9].

## 3. Results

### 3.1. 16S rRNA Gene Sequencing Outputs

Illumina MiSeq produced a total of 19.6 million paired-end reads (min = 706, max = 706,327, median = 46,818 per sample) from a total of 245 samples. Three samples with low quality scores were excluded from alpha and beta diversity analyses. Two hundred and forty-two samples that passed the quality check comprised 11 million total features (min = 254, max = 392,829, and median = 27,688 per sample) and 7511 unique features (i.e., amplicon sequence variant (ASVs)). Rarefaction curves for all considered alpha diversity metrics (i.e., observed OTUs, Chao1, Faith’s phylogenetic diversity, and Shannon and Simpson indices) showed that the sequencing depth was sufficient to reveal rare bacterial taxa. For the computation of alpha and beta diversity metrics and other bioinformatic analyses, the data were normalized to 7790 reads per sample using rarefaction.

Calves in the sick groups, those infected with *M. haemolytica* before the enrofloxacin treatment, demonstrated mild signs of respiratory disease. However, the diversity metrics did not show significant differences between the sick and the healthy groups; thus, these groups were collapsed together into low and high dose treatment groups based on the dosing regimens of enrofloxacin. Hence, the focus of this data analysis was on differences in dosing regimens (i.e., control versus low dose versus high dose) and samples collected before and after enrofloxacin injection (i.e., pre-treatment versus post-treatment). The comparisons were made at different levels: between treatment (enrofloxacin injected combined groups, now onward treatment group) and control, between pre-and post-treatment samples for the combined treatment group, between pre-and post-treatment samples of the high and the low dose groups separately, and among sampling days (sampling day right before enrofloxacin injection on Day 21 with the subsequent sampling days).

### 3.2. Microbial Profiling

#### 3.2.1. Subsubsection Alpha and Beta Diversity Metrics Show a Significant Microbial Shift following Enrofloxacin Administration

To compare the difference in the microbial richness and evenness between groups, observed OTUs and the Shannon index were computed using Qiime2, respectively. However, none of these comparisons between the calf groups (control, low dose, and high dose, *p* > 0.05) were found to be statistically significant. The line graph in Figure 1 compares alpha diversities among the control, the low dose, and the high dose groups on each sampling day. The low and high dose groups were collapsed together (i.e., combined treatment group) to evaluate differences in diversity metrics between pre-and post-treatment samples. For each metric, the average of the four pre-treatment sampling days and the average of the three post-treatment sampling days were computed and compared. Accordingly, the microbial richness significantly increased following the treatment (observed OTUs, mean: pre = 279, post = 325 [adj. *p* = 0.033]), and the microbial evenness also increased marginally after the treatment (Shannon index, pre = 5.8, post = 6.2 [*p* = 0.047, adj. *p* = 0.071]). Consistent with our hypothesis, microbial diversities did not differ significantly between the control group (including all sampling days) and the combined pre-treatment samples (observed OTUs: control = 278, pre = 279, and Shannon index: control = 5.7, pre = 5.8, *p* > 0.05 for both). The effect of enrofloxacin on microbial richness did not differ between the low and high dose groups (Figure 1). Microbial evenness was significantly lower on sampling day 22 compared with day 24 in the low dose group (Figure 1).

Further analysis was conducted to check the difference among post-treatment sampling days (Days 22, 24, 28) with the immediate pre-treatment day (Day 21). In the combined treatment group, the observed OTUs and the Shannon index on sampling day 22 (i.e., 24 h after enrofloxacin administration) were significantly lower than sampling day 24, which suggested alterations in the microbial richness and evenness due to the treatment. Furthermore, we used linear mixed-effects models in QIIME 2 to assess the effects of sampling days on microbial diversities. Based on this analysis, the effects of sampling days on the microbial richness and evenness were not significant for the control group (*p* > 0.05) and marginally significant for the combined treatment group (*p* = 0.08). Moreover, a Kruskal–Wallis test was run to assess differences of alpha diversities by sampling days between the control and treatment samples. Accordingly, none of the sampling days were significantly different for observed OTUs and Shannon index in the control group. In contrast, they were significantly different in the combined treatment group, indicating that the changes in the microbial diversities were driven by enrofloxacin injection rather than by age (Figure 2).

To evaluate the difference in beta diversity between pre- and post-treatment microbial communities, as well as between the combined treatment and control groups, we computed a Bray–Curtis dissimilarity index using Qiime2. When the sampling timepoints collapsed to pre- and post-treatment, the difference between pre- and post-treatment samples was significantly different (Table 3, *p* = 0.001). Furthermore, the principal coordinate analysis indicated a clustering difference between pre- and post-treatment samples (Figure 3).

#### 3.2.2. Relative Abundances of Certain Bacterial Taxa Varied Significantly between Pre- and Post-Treatment Samples

The average relative abundances of phyla in pre- and post-treatment samples were compared using the Wilcoxon rank-sum test. Twenty-six phyla were detected in our samples, with *Firmicutes*, *Bacteroides*, and *Proteobacteria* predominating in both pre- and post-treatment samples, but their proportions were not significantly altered following enrofloxacin injection (Figure 4). In contrast, *Spirochaetes* (pre-treatment average relative abundance 6.05%, post-treatment 8.48%), *Tenericutes* (pre-treatment 1.74%, post-treatment 2.30%), *Verucomicrobia* (pre-treatment 0.72%, post-treatment 1.10%), and *Kiritimatiellaeota* (pre-treatment 0.11%, post-treatment 0.16%) significantly increased in relative abundance after the enrofloxacin treatment, while *Euryarchaeota* (pre-treatment 1.40%, post-treatment 0.48%), *Cyanobacteria* (pre-treatment 0.63%, post-treatment 0.30%), and *Actinobacteria* (pre-treatment 0.32%, post-treatment 0.19%) decreased significantly (*p* < 0.05) after enrofloxacin injection. Appendix A depicts the relative abundances of the top ten phyla by groups and sampling days.

A total of 442 genera were identified in this study; however, 183 of them were found in less than 10% of the samples; thus, they were excluded from statistical analyses. Eighteen of them are members of the core fecal microbiota, existing in ≥90% of the samples [26] (Appendix A). Comparisons of genus count per sampling day between the treatment and control groups did not significantly differ (Figure 5). This study paid particular attention to the genus *Campylobacter* due to its public health importance and rising fluoroquinolone resistance among its strains. However, this genus was detected in 53 of the 245 total samples, most of which were the pre-treatment samples (*n* = 37, only seven in the post-treatment samples). Nine of them were detected in the samples from the control calves. Interestingly, prior to enrofloxacin administration on Day 21, *Campylobacter* was detected in six calves in the treatment groups; however, 24 h and 72 h after the treatment, only one calf was positive each day. On day 28, however, five calves turned *Campylobacter* positive (four in the high dose group and one in the low dose group), suggesting the emergence of resistant strains. The relatively lower detection level of *Campylobacter* in the two post-treatment sampling days than the pre-treatment samples might also suggest that enrofloxacin curtailed the shedding of this organism in the calf feces.

#### 3.2.3. Analysis of Composition of Microbiomes (ANCOM) Shows Disruptions of Certain Bacterial Taxa by Enrofloxacin Injection

ANCOM was used to identify bacterial taxa that showed significant differences in abundance levels between pre- and post-treatment samples. Thirty-five percent (53/152) of the identified bacteria families were significantly different between pre- and post-treatment samples. Families such as *Bifidobacteriaceae*, *Eubacteriaceae*, *Akkaermansiaceae*, *Proteobacteria*-uncultured, *Enterobacteriaceae* decreased in relative abundance, whereas *Bacteroidales*_UCG-001, *Prevotellaceae*, and *Tenericutes*- gut_metagenome increased in relative abundance following treatment. The top seventeen altered families are presented in Table 4. The ANCOM analysis was also performed at the genus level, where a number of genera that showed significant compositional changes after the treatment were identified (Appendix A). Since this study was conducted using young calves (three to four months old at the enrollment) over 28 days, we anticipated that the microbial changes could occur due to age. Thus, to minimize the effects of time on microbial variations, a separate ANCOM analysis was conducted for the last week of their life on the animal research facility (i.e., sampling days 21, 22, 24, and 28). We found that three genera significantly increased (*Ruminococcaceae*_UCG-010, *Rhodospirillales*_uncultured, and *Turicibacter*), but only one genus significantly decreased (*Phascolarctobacterium*). Comparisons of the last three sampling days (post-treatment) with the samples collected right before enrofloxacin administration (Day 21) showed significant variations in genera between sampling days. Table 5 presents the comparisons of six genera that showed significant variations in the last three sampling days compared to fecal samples taken on the enrofloxacin injection day (Day 21). Furthermore, ANCOM analysis was carried out to check if the dosing regimens might have affected gut microbiota differently at the genus level. Strikingly, only two genera were significantly affected by the low dose of enrofloxacin compared to four genera in the high dose group (Table 6). Both genera increased following the treatment in the low dose group, while two of them increased, and the other two decreased in the high dose group. Genus *Prevotellaceae*_UCG-003, which increased post-treatment in the high dose, was a member of the core fecal microbiota.

### 3.3. Metagenomic Hi-C Results Show Changes in Copy Numbers and Host Ranges of ARGs following Enrofloxacin Administration

Pooled fecal samples from four groups of calves were processed using the metagenomic proximity ligation approach (ProxiMeta Hi-C) to assess the effect of enrofloxacin on ARG dynamics. Thirty-two genes conferring resistance to six classes of antibiotics were detected in the samples. Tetracycline, macrolide, and aminoglycoside resistance genes were highly abundant in both pre- and post-treatment pooled samples in all groups, whereas phenicol and sulfonamide resistance determinants were the least abundant. Table 7 presents the number of hits and number of bacterial clusters carrying resistance genes for these six classes of antibiotics in each calf group. The copy number and the host ranges of the ARGs were dynamic in both the control and the treatment groups; however, these variations were more prominent in the treatment groups, and the low dose induced a remarkable reduction in the number of hits and host ranges (Table 8 and Figure 6). Out of the 32 total unique ARGs detected, 5, 4, 7, and 9 increased, and 12, 18, 12, and 10 decreased notably in the number of hits in the control, low dose, high dose healthy, and high dose sick groups, respectively.

Similarly, the enrofloxacin treatment seemed to affect the number of bacterial hosts identified carrying the resistance markers remarkably. In the treatment groups, the host ranges of 21 ARGs were altered (either decreased or increased after the treatment) compared to 14 ARGs that varied in the control group. Generally, there was no remarkable difference between the two high dose groups. In contrast, it appears that high dose enrofloxacin induces the increased copy number and host ranges of ARGs, whereas the low dose induces a reduction of the number of hits and host ranges of ARGs.

Five types of aminoglycoside resistance genes were detected in the current study, which included *aph*2, *aph*3, *ant*6, *ant*9, and *sat*. The *aph*2 and *aph*3 genes were carried frequently on the same contigs. The number of hits and clusters hosting them showed variations in all groups; however, the low dose healthy group showed apparent differences between pre- and post-treatment pooled samples. All these genes declined both in the copy number and host range after the treatment; for instance, *aph*2 decreased from 141 and 23 to 7 and 6, and *ant*6 decreased from 193 and 33 to 3 and 2 in copy and host numbers, respectively. Aminoglycoside resistance genes were associated with *Prevotella* and *Bacteroides* species in *Bacteroidetes* and *Clostridiales* species in *Firmicutes* (Appendix A). Furthermore, several bacterial taxa, including *Firmicutes* and *Subdoligranulum* species, acted as the reservoirs of *sat.*

Resistance genes that confer resistance against β-lactam antibiotics were also detected in the current study. However, only two of them showed notable variations. The *rob* gene significantly increased from 12 to 63 hits, with an increase of one host range in the low dose group, whereas 162 hits of the *oxa* gene were detected in four bacterial clusters in the post-treatment sample, but absent in the pre-treatment sample in the high dose healthy group. Unclassified *Firmicutes*, *Prevotella,* and *Treponema* species were associated with the carriage of the β-lactam resistance genes (Appendix A).

A number of macrolide resistance genes were identified in the pooled fecal samples (Table 7). High variations in copy numbers and host ranges were more pronounced in the three treatment groups than in the control group. In the low and high dose healthy groups, most of the ARGs were affected alike; the host ranges reduced after the treatment in these two groups. In contrast, copy number and host ranges increased for most of these ARGs in the high dose BRD group; for instance, *erm*Q and *mef*E increased respectively from 2 to 87 and from 1752 to 4693 in the number of hits, and from 1 to 12 and from 5 to 45 in the host ranges. Moreover, *mef*E was the most abundant in the number of copies and host ranges. Macrolide resistance genes were mainly carried by *Bacteroides*, such as *Prevotella* and *Bifidobacterium* species (Appendix A).

A few resistance genes conferring resistance to phenicol and sulfonamide were reported in this study. *cf**r* remarkably reduced in the host range in all treatment groups compared to the control and decreased in the copy numbers in low dose and high dose healthy groups, unlike the control and high dose BRD groups following enrofloxacin administration. The phenicol resistance genes were hosted by *Bacteroides* species, *Prevotella* species, *Treponema* species, *Mycoplasma* species, and *Bifidobacterium pseudolongum* (Appendix A).

Twelve types of tetracycline resistance genes were detected in the current study. Interestingly, a number of these genes showed a significant variation between the pre- and post-treatment samples in the three treatment groups compared to the control group. For instance, *tet*O decreased in the number of copies with no remarkable changes in the host range following the treatment in all treatment groups. In contrast, *tet*W significantly decreased in the copy number in the treatment groups compared to the control. Strikingly, *tet*Q increased in the copy number and host range only in the high dose BRD group, while these values decreased in both treatment groups (low dose and high dose healthy) and the control alike. The tetracycline resistance genes were hosted by several bacterial taxa; notably, *tet*W was the most widely distributed ARG with its presence in most of the bacterial taxa reported in this study (Appendix A).

### 3.4. Quantitative Alterations in Selected Resistance Determinants

Five antibiotic determinant genes were selected to assess the quantitative variation induced by enrofloxacin treatment in three calf groups and to validate the Hi-C-based quantitative evaluation (Table 9). Among them, *tet*W was significantly increased in the low dose healthy group compared to the control group. This finding is inconsistent with the Hi-C metagenomic results, in which 721 hits were recorded in the pre-treatment pooled sample, while only 186 hits were observed in the post-treatment sample. The *erm*B gene also significantly changed in the high dose healthy group, and a similar change was observed in the Hi-C results. The other three ARG types (*tet*O, *tet*X, and *erm*F) did not show any significant differences between the pre- and post-treatment pooled samples in either the low or high dose treatment groups, as measured by qPCR. Except for *tet*O, the Hi-C results for the other two ARGs (*tet*X and *erm*F) looked consistent with the qPCR readings.

## 4. Discussion

Enrofloxacin is an effective short-acting fluoroquinolone drug used to prevent and control bovine respiratory disease [27,28]. Its use in livestock potentially contributes to the alteration of gut microbiota and resistome in animals. Enrofloxacin is recommended in either multiple or single doses. The single therapeutic injection can be administered in a wide range of dosages (7.5–12.5 mg/kg) [15], providing an opportunity to explore whether or not the upper and lower dose limits distinctively affect the microbiota and antibiotic resistance. In this study, the relative effects of the upper limit (12.5 mg/kg) and the lower limit (7.5 mg/kg) of a single therapeutic dose of enrofloxacin on calf fecal microbiota and resistome were assessed. In addition to comparing the high and low doses, the data of the two dosage groups were merged and then compared to the control group, to determine the extent of microbial alterations attributed to enrofloxacin. Accordingly, a single dose of enrofloxacin administered subcutaneously to calves affected microbial compositions and diversities significantly compared to the control group, consistent with a previous study [5]. Comparisons of the upper and lower dose limits also demonstrated that the two dosages affected compositions of certain bacterial taxa differently; a more significant number of bacterial genera were shifted in the high dose group than the low dose group. In addition, the resistome profiles were significantly affected in calf groups that received enrofloxacin compared to the control group.

Enrofloxacin altered the richness and evenness of gut microbiota significantly; both metrics were lower at the 24 h post-treatment samples, compared to the 72 h post-treatment samples when the data of the high and the low dose groups were combined. This was expected, since the active concentration of enrofloxacin and ciprofloxacin reaches its maximum concentration in the intestine around 24 h post-inoculation, as reported in a previous study [19]. When examining the high dose group (pre-treatment versus post-treatment) and the low dose group (pre-treatment versus post-treatment) separately, there was no significant change in microbial richness between pre- and post-treatment samples in either group. However, microbial evenness was significantly altered in the post-treatment samples (Days 22 versus 24, *p* = 0.008, adj. *p* = 0.068) of the low dose group but not the high dose group (*p* > 0.05). Furthermore, microbial diversities between groups measured by the beta-diversity metric (i.e., Bray–Curtis) have indicated a significant difference between pre-and post-treatment microbial community structures, as well as a significant difference between the high dose and the low dose groups.

Gut microbiota highly fluctuate in the early life of calves until weaning, after which adult-like microbiota is established [2]. The age of calves at enrollment in this study was between 12 and 16 weeks; thus, significant microbial variation related to early life was not expected. Moreover, the effects of age were found to be insignificant, as shown by the linear mixed model and the Kruskal–Wallis test in this study. Therefore, the variations of gut microbiota between sampling days observed in this study are more likely to be attributed to the effects of enrofloxacin. This is contrary to a study conducted by Holman et al. [29] in steers, where time, rather than antibiotic administration, was reported as the main driver of changes in microbial community structures. Holman et al. [29] followed the animals for 34 days post-treatment, unlike in the present research, where the study was terminated seven days after the treatment. Similar to the present study, Holman et al. [29] also observed a significant variation of microbial diversities between pre-medication and immediate post-medication days (i.e., sampling days two and five). Their findings imply that the effects of antibiotics are reversed in the long run, possibly due to the repopulation of the intestine by microorganisms from the forestomach of cattle [19]. However, due to the shorter duration of our follow-up, we were unable to determine if these values returned to normal on their own over a more extended time.

The proportions of roughly one-third of the identified bacterial families and several genera were affected by the treatment (Table 2); some of these families decreased, and others increased significantly after enrofloxacin administration. The alteration of bacteria following enrofloxacin injection in the current study is consistent with the recent report of Ferguson et al. [19]. They also reported the changed frequency of *Prevotellaceae*, *Christensellaceae*, *Erysipelothrichaceae*, and *Enterobacteriaceae* 48 h after enrofloxacin injection to steers. In the present study, one notable change consistent with the report of Ferguson et al. [19] at the genus level is that frequency of *Escherichia-Shigella* significantly decreased at 24 h and 72 h post-enrofloxacin injection in the treatment group compared to the control group. Additionally, the number of calves carrying *Campylobacter* reduced remarkably after the treatment. This was demonstrated in a mouse study as well, where subcutaneously administered ciprofloxacin reduced *Proteobacteria* species in feces to below the detection limit, with minor effects on other phyla [30].

The high dose enrofloxacin resulted in the shifting of a more significant number of genera than the low dose, which was four genera versus two genera, respectively (Table 4). This result is consistent with our hypothesis that dosing would further impact microbial community distributions due to a high dose resulting in a higher concentration of enrofloxacin and its metabolite in the intestine than the low dose. The high dose and the subsequent higher concentration of drug in the intestine have a more prominent effect on the gut microbiota than the small dose, because the period of time for which the concentration of drug remains above the minimum inhibitory concentration (MIC) is greater in the former case [31]. In the current study, the proportion of two of the four genera shifted by the high dose was significantly decreased (Table 4). Our observations suggest that the high dose is lethal, while the low dose may favor the emergence of resistant organisms and their clonal expansion. This phenomenon might be best explained by the concept of a mutant selection window. It is defined as the drug concentration between the MIC of a susceptible strain and the MIC of a resistant strain. A concentration above this window is no more capable of selecting for resistant strains; however, the drug becomes lethal to gut microbes [31,32,33].

Overall, we have seen a transient alteration of fecal microbiota following enrofloxacin administration to beef calves in the current study. The reason that enrofloxacin caused only a temporary microbial shift might be due to the repopulation of intestinal microbiota by forestomach organisms [19,26,29], which is contrary to monogastric animals, including humans, where the effects of antibiotics on gut microbiota are relatively protracted [34,35]. A study that characterized microbiota across the digestive tract of a steer revealed distinctive patterns in microbial diversity and relative abundance among rumen, small and large intestines; nevertheless, a large number of OTUs were found to be shared among different segments of the tract [36]. In the current study, we did not evaluate the similarity between the microbiota of feces and forestomach to substantiate the repopulation hypothesis.

The transient alteration of intestinal microbiota may suggest that antibiotics might not lead to severe microbial dysbiosis that compromises the growth of animals and health in ruminants; however, antibiotics may exercise intense selection pressure that can lead to the enrichment of resistant bacterial strains and antibiotic determinants. Studies have demonstrated an increased prevalence of resistant bacterial strains being shed in feces following antibiotic administration in cattle, even when they are administered at subtherapeutic amounts [37,38]. Thus, while gut microbiota are resilient to antibiotics in the long term, it should be noted that the selective pressure by antimicrobials may lead to the increased shedding of resistant bacteria and genes in cattle. This situation entails a risk of contaminating the environment and poses a public health threat; thus, it necessitates critical analysis when making treatment decisions.

In the present study, the impacts of enrofloxacin on the fecal resistome profiles were assessed using the metagenomic proximity ligation (Hi-C) approach. This novel method identified 32 unique ARGs that confer resistance against six antibiotic classes. However, no antibiotics from the classes identified were administered during the study. Thus, the alteration of their resistance genes following the administration of an unrelated antibiotic (enrofloxacin) might indicate the co-selection or co-transfer of these genes [39,40,41]. Some of the ARGs, such as *aph*3 (aminoglycoside resistance gene) and *rob* (β-lactam resistant gene), were found on the same contigs; their co-localization suggests the possibility of co-selection and co-transfer to a new host. Tetracycline resistance genes were more abundant than the resistance genes of any other antibiotic class, which corresponds to the wider use of tetracycline antibiotics on cattle farms, as well as the existence of a larger number of tetracycline ARG types compared to other antibiotic classes [42,43,44]. The lower hit numbers of the ARGs of other antibiotics, such as sulfonamides, which are commonly used on beef farms, might be associated with reduced use of these antibiotics on the calf source farm. A study conducted in Canada revealed that the abundance of sulfonamide resistance genes reflected their use on farms; farms that did not have a recent history of using these antibiotics had a low abundance of their resistance genes [42]. In the current study, however, the history of antibiotic uses on the farm from which the study calves originated was unknown. Nevertheless, these calves were not administered any antibiotics before they were enrolled in this study.

The effects of the enrofloxacin on the emergence and spread of β-lactam, phenicol, and sulfonamide did not appear significant. However, a few ARGs, such as *oxa* (a β-lactam resistance gene) and *cm*X (a phenicol resistance gene), were identified following the administration of high dose enrofloxacin. One hundred and sixty-two hits of *oxa* were detected in four unclassified clusters of *Firmicutes*. There are different variants of the *oxa* gene; some of them encode β-lactamase and carbapenemase enzymes that confer penicillin and carbapenem resistance in *Actinobacterium* and *Enterobacteriaceae* species [45,46,47,48]. In this study, however, the specific variants present in the fecal samples were unknown. *cm*X was detected in the high dose BRD groups in two bacterial species: three hits in *Prevotella stercorea* and four hits in *Bifidobacterium pseudolongum* subsp. Globosum. This gene confers resistance by exporting chloramphenicol out of the cell. The latter species is thought to be the origin of *cm*X and can transfer it into *Enterobacteriaceae* species with public health importance [49]. Furthermore, the *cfr* gene that encodes rRNA methylases and mediates phenicol resistance by target site modification was detected in several bacteria in all groups. The reservoirs of this gene included *Prevotella*, *Treponema*, *Mycoplasma*, and *Bacteroides* species. Previously, it was detected in *Bacillus*, *Enterobacter*, *Staphylococcus*, *Streptococcus*, and other species [50]. Additionally, five different ARG types rendering resistance to aminoglycoside were detected. The aminoglycoside-resistant genes were dynamic in all groups; however, their host ranges were reduced in the low dose healthy and high dose sick groups, while increasing in the high dose healthy group.

There are noticeable impacts of the enrofloxacin treatment on macrolide resistance genes in the present study. The copy numbers and the host ranges were altered remarkably following enrofloxacin administration in the treatment groups compared to the control group. Some of the ARG hosts were affected by the treatment, as they decreased after the treatment, especially in the low and high dose healthy groups. These hosts might be susceptible to enrofloxacin and/or its metabolite, ciprofloxacin. The most abundant macrolide ARG was *mef*E, hosted by several bacterial taxa. *Mef*E provides a macrolide-efflux system in a wide range of bacterial species, including *Enterococcus*, *Staphylococcus,* and *Streptococcus* species [51].

Twelve tetracycline resistance determinants have been detected in the present study. Comparisons between the control and the treatment groups show that the types of tetracycline resistance genes varied highly between the pre- and post-treatment pooled samples in the latter groups. In the control group, eight and seven types of tetracycline resistance genes were detected, whereas eight and ten in the low dose healthy and ten and five in the high dose healthy group were identified in pre- and post-treatment samples, respectively. Given that tetracyclines are the most commonly used antibiotics in livestock, the present findings are not surprising [42,43]. Tetracycline resistance genes were observed in calves as young as two weeks old, with no prior antibiotic exposure [52]. However, the high variations in the treatment groups suggested that enrofloxacin affected the resistome profiles, possibly in two ways. (1) The treatment might promote the transfer of resistance genes between bacterial taxa, an established phenomenon in a complex microbial community, such as the gut microbiota [53,54]. In this study, the host ranges of three and five ARGs have increased following the treatment, respectively, in the high dose healthy and sick groups. (2) Its selective pressure might affect the abundance of the reservoirs and, thus, affect the abundance of these ARGs. In this study, the tetracycline ARGs were mainly carried by the core fecal microbiota (Appendix A), such as *Bacteroides*, *Prevotella*, and *Treponema* species. Their abundance increased significantly following the enrofloxacin administration in the treatment groups (Appendix A).

In the current animal study, there is some evidence which shows that enrofloxacin treatment enhanced the emergence and/or spread of ARGs. The first evidence is that the type of ARGs (those of tetracyclines and macrolides) varied more between pre- and post-treatment in the treatment groups than in control. A larger number of new ARGs was identified following the treatment in the treatment groups than the same sampling day in the control. Second, the number of hits for certain ARGs significantly altered in the treatment groups compared to the control. For instance, the number of hits for *tet*40, *tet*O, and *tet*W showed significant variations in the treatment groups. Thirdly, the number of clusters carrying certain ARGs showed a significant variation in the treatment groups. For instance, *ant*6 was detected in 33, 15 (equivalent to 10 when normalized), and 6 in the pre-treatment sample, while these numbers altered to 2, 14, and 18 in the post-treatment in the low dose, high dose healthy, and high dose BRD groups, respectively. However, there was no change for the *ant*6 host range in the control group. This may suggest the occurrence of horizontal ARG transfer among bacterial taxa potentiated by the treatment. Such variations were not observed in the pooled samples from the control calves, which received no antibiotics during the course of the study.

In the present study, the resistome profiles have been affected differently depending on the dose of enrofloxacin. Much larger numbers of ARGs have been reduced in copy numbers and host ranges in the low dose group than in the high dose group; conversely, a more significant number of ARGs increased in the high dose group after the treatment (Figure 6). The high dose results in a higher concentration of enrofloxacin and its metabolite in the system, which means that the clearance time is extended [55]. Studies have shown that enrofloxacin and its metabolites are excreted from the plasma to the intestinal lumen through the active transport system, as well as enterohepatic circulation accumulates them in the intestine [56,57]. Furthermore, their concentrations increase from the proximal to the distal part of the intestine. This implies that a higher dose resulting in a higher concentration in the blood circulation that leads to a higher concentration for a more extended time in the intestinal lumen. Thus, the higher dose induces selective pressure for a longer time than the low dose, favoring the emergence and proliferation of resistant strains. The extended selective pressure also provides a fitness advantage to the resistant strains to predominate over the susceptible strains and favors the transfer of ARGs horizontally between bacteria. In the case of the low dose, since the drug clearance is relatively quick and the selective pressure wanes fast, the fitness cost of the resistant strains is high, and thus, they are outcompeted and predominated by susceptible strains when the selection pressure is not present anymore. Historically, it has been viewed that the highest tolerable dose is effective in minimizing the emergence of antimicrobial-resistant strains. However, recent studies have revealed that the lowest possible, yet clinically effective dose might better minimize the emergence and maintenance of antimicrobial resistance in complex microbial communities [58,59].

The resistome profiles were affected similarly in the two high dose groups, implying that the *M. haemolytica* challenge did not cause a significant difference. Nevertheless, a few ARGs have responded to the treatment differently; for instance, the host ranges of aminoglycoside ARGs increased in the healthy group, while they decreased in the sick group following the high dose administration. Studies have shown that infections affect the pharmacokinetics and pharmacodynamics of drugs [60,61]. For instance, the elimination rate of danofloxacin was slower in *Pasteurella multocida*-infected ducks than in healthy ducks [60]. This means slow elimination results in gut microbial exposure to this antibiotic and its metabolite for an extended time, resulting in clonal expansions of ARG hosts and their transfer among bacterial species. However, in the current study, we did not see these effects, which might be attributed to one or both of the following reasons. First, in the current study, the calves challenged with *M. haemolytica* did not exhibit typical signs of bovine respiratory disease, meaning that the severity of the infection might not have been strong enough to alter the pharmacokinetics and pharmacodynamics of enrofloxacin and ciprofloxacin. Second, enrofloxacin was administered eight days (Day 21) after the M. haemolytica challenge (Day 13), when all moderate respiratory signs had already waned, and the post-treatment samples used for these comparisons were collected as much as two weeks (Day 28) after the challenge.

In this study, the Hi-C method enabled us to know the bacterial taxa that carry each identified ARG based on the proximity ligations. Consistent with previous studies [62], most ARGs were carried by species in *Bacteroidetes* and *Firmicutes* phyla. *Bacteroides*, *Clostridium, Prevotella*, and *Treponema* species have been found to host most of these ARGs. These findings provide insight into the origin and dissemination of ARGs, which are crucial to successfully combat the emergence and spread of antimicrobial-resistant bacterial strains and genes [49]. Unfortunately, the cost of Hi-C sequencing limited the number of samples that we were able to compare using this technology. Combined with differences in sequencing depth, this results in low sample numbers and variability in data. We have tried to control for sequence depth by normalizing the data comparison to the number of bacterial clusters identified; however, the results of this study still need to be interpreted with these limitations in mind. It is clear that this technology allows us to investigate complex resistome with new insights and future work to allow for better normalization methods, and the ability to use the Hi-C to inform targeted selective sequencing applied to larger sample sizes will greatly benefit the outcomes of these studies.

Comparison of the pattern of ARG quantitative changes between the qPCR method and the Hi-C method showed a good agreement overall. Out of ten qPCR measurements, six of them were consistent with the Hi-C results. Based on the qPCR measurements, *tet*O did not have significant quantitative changes, whereas the Hi-C result indicated some reduction following enrofloxacin injection. Comparison of the quantitative changes of the ARGs is more reliable using qPCR than the Hi-C method, because qPCR readings were normalized using the 16S rRNA gene. Nevertheless, the Hi-C method has multiple advantages: it is less prone to bias, since primers are not required [63]; bacterial taxa associated with ARGs can be identified; it provides insight into the spread of ARGs [64].

A number of the ARGs observed in this study can confer resistance to medically important antimicrobials of humans and animals. Interestingly, treatment with enrofloxacin was associated with both increased and decreased ARG hits, as well as both increased and decreased host range observed to be carrying the ARGs. As evidenced in Figure 6, these dynamics occur even in control animals not exposed to antibiotic treatment, and in some cases, (low dose) antibiotic treatment resulted in a net decreased abundance and host range of ARGs. Importantly, the majority of taxa observed to have ARGs in this study are not direct human or animal pathogens, and hence pose limited direct risk of clinically significant treatment failure in a human or animal. However, as demonstrated by this study, the resistome dynamics, both in treated and untreated (control) animals, are complex and require additional research to understand the drivers and impacts of ARG transmission. Understanding ARG disseminations and minimizing their detrimental impacts require concerted efforts among animal, human, and environmental health expertise; in short, a “One Health” approach [6,7,8].

## 5. Conclusions

It can be concluded from the current study that a subcutaneously injected single dose of enrofloxacin alters calf fecal microbiota and resistome profile significantly. Comparisons of the upper and lower dose limits of the drug show some differences between the two doses in affecting microbial diversities and compositions. The high dose appears to have more impact than the lower dose at the level of bacterial taxa. Similarly, the two doses affected the resistome profiles differently. The high dose of enrofloxacin appeared to induce clonal expansion and potentially the transfer of ARGs among gut microbial species. Finally, the *M. haemolytica* challenge was not found to affect the fecal microbiota and resistome profile, which might be due to our inability to induce typical BRD signs and the delay between the challenge and the antibiotic administration when the BRD signs are already subsidized.

## Figures and Tables

**Figure 1 microorganisms-09-02162-f001:**
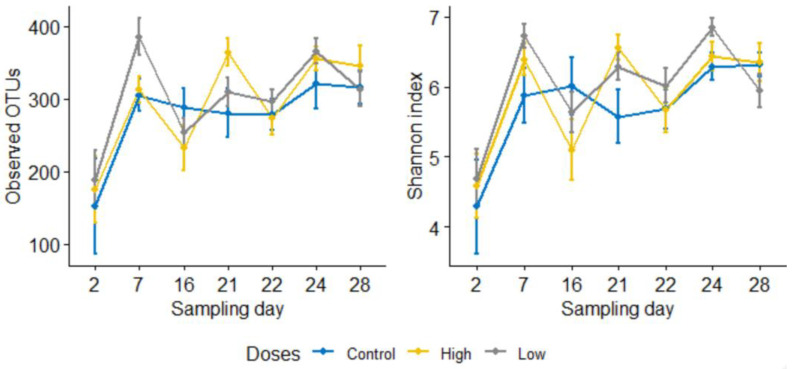
Line graphs comparing microbial richness and evenness measured by observed OTUs (**left**) and Shannon index (**right**) in the control group and combined low and high dose enrofloxacin treatment groups by sampling days. The differences were insignificant among the three groups on each sampling day for both metrics (*p* > 0.05). All calves were inoculated with laboratory strains of *C. jejuni* on day 5; the BRD groups were challenged with *M. hemolytica* on day 13; and the treatment groups were administered enrofloxacin on day 21.

**Figure 2 microorganisms-09-02162-f002:**
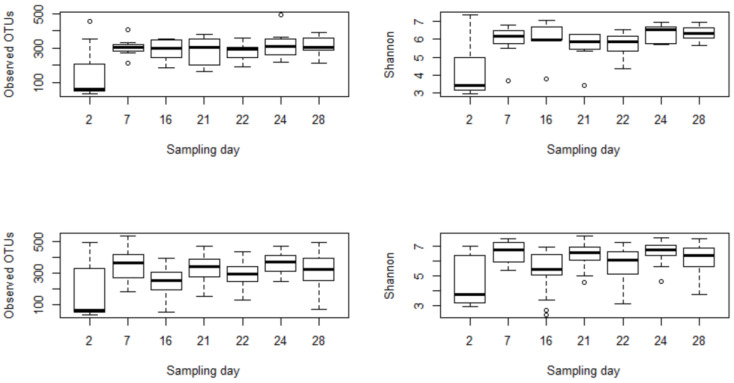
Comparisons of microbial richness (observed OTUs) and evenness (Shannon index) between the control group (**top**) and the combined treatment group (**bottom**) by sampling days. The low and high dose groups were collapsed together into a single treatment group to assess the effects of sampling days on microbial diversities. The difference among sampling days was insignificant in control (*p* > 0.05), but it was significant in the treatment group (*p* < 0.05). All calves were inoculated with laboratory strains of *C. jejuni* on day 5; the BRD groups were challenged with *M. hemolytica* on day 13; and the treatment groups were administered enrofloxacin on day 21.

**Figure 3 microorganisms-09-02162-f003:**
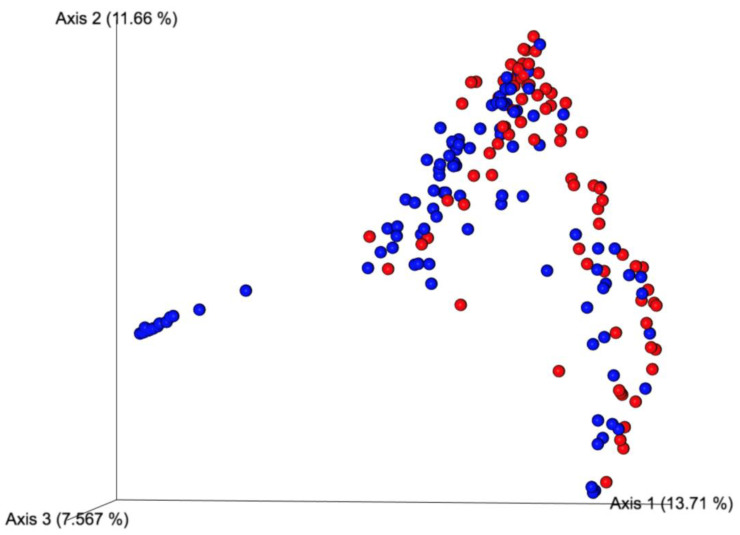
Principal coordinate analysis showing a subtle difference microbial clustering between pre- (blue dots) and post-treatment (red dots) samples in the combined treatment groups.

**Figure 4 microorganisms-09-02162-f004:**
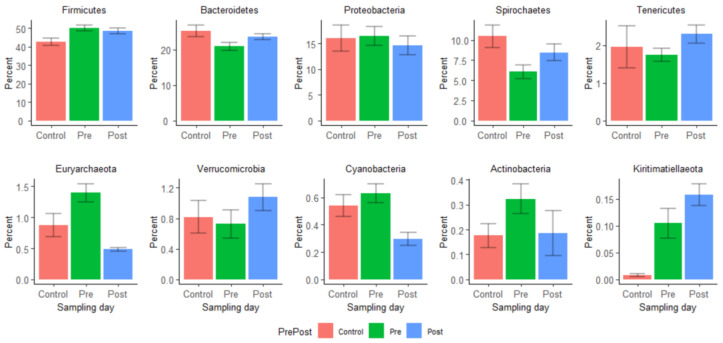
The relative abundance of the ten most frequent phyla was compared between pre- and post-treatment and control samples.

**Figure 5 microorganisms-09-02162-f005:**
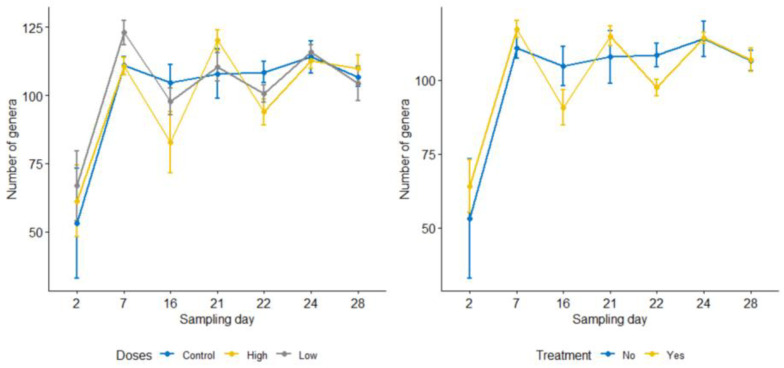
Comparisons of bacterial genus counts by sampling days among control, low dose, and high dose groups (**left**) and between the control and the combined treatment groups (**right**). The differences between groups on each sampling day were not significant for both comparisons (*p* > 0.05).

**Figure 6 microorganisms-09-02162-f006:**
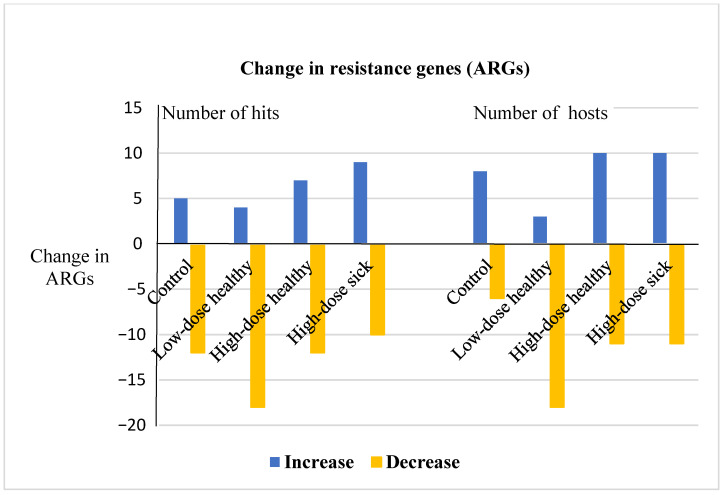
Changes in the copy numbers and the host ranges of ARGs in four groups of the study calves. At least an increase or decrease by 25% was considered in the number of hit changes, while an increase or decrease of one cluster for ARGs with at least two copies was considered for the host range changes. The numbers of clusters were different between pre- and post-treatment samples, but they were normalized to make a reasonable comparison.

**Table 1 microorganisms-09-02162-t001:** Summaries of activities performed during 28 days of the study course.

Activities		Pre-Treatment				Post-Treatment				Involved Group *
	Day 0	4	7	13	16	21	22	24	28	
1. Calf examination, weighing, and room assignment										All groups
2. *C. jejuni* inoculation										All
3. *M.* *haemolytica* inoculation										Low and high- dose BRD groups
4. Enrofloxacin injection										All, except the control
5. Fecal sample collection										All
6. Lung examination										All

* Five groups (each consisting of seven calves): control, healthy-low dose, BRD/sick-low dose, healthy-high dose, and BRD/sick-high dose groups. Each shaded box indicates the day that the respective activity was conducted.

**Table 2 microorganisms-09-02162-t002:** Primers used to quantify selected resistance genes in calf pooled fecal samples using qPCR.

ARGs	Forward Primer	Reverse Primer	Reference
*erm*B	TGAAAGCCATGCGTCTGACA	CCCTAGTGTTCGGTGAATATCCA	Looft et al., 2012 [9]
*erm*F	TTTCAAAGTGGTGTCAAATATTCCTT	GGACAATGGAACCTCCCAGAA	
*tet*O	ATGTGGATACTACAACGCATGAGATT	TGCCTCCACATGATATTTTTCCT	
*tet*W	TCCTTCCAGTGGCACAGATGT	GCCCCATCTAAAACAGCCAAA	
*tet*X	AAATTTGTTACCGACACGGAAGTT	CATAGCTGAAAAAATCCAGGACAGTT	

**Table 3 microorganisms-09-02162-t003:** Comparisons of Bray–Curtis dissimilarity index between pre- and post-treatment samples in the treatment groups.

	Comparisons	Bray–Curtis (Adjusted *p*-Value)
1	Low dose: pre- vs. post-treatment	0.002
2	High dose: pre- vs. post-treatment	0.002
3	Low dose pre-treatment vs. high dose pre-treatment	0.272
4	Low dose post-treatment vs. high dose post-treatment	0.003
5	Pre- and post-treatment (low and high dose groups combined)	0.001

**Table 4 microorganisms-09-02162-t004:** Results of ANCOM analysis showing bacterial families significantly different among control, pre-treatment, and post-treatment samples. High W values indicate significant differences in abundance levels between groups.

	Families	Control vs. Pre vs. Post (W)	Pre vs. Post (W)	Pre vs. Post (Change)
1	Bacteroidetes_Bacteroidia_Bacteroidales_uncultured	153	143	Increased
2	Proteobacteria_Gammaproteobacteria_Enterobacteriales_Enterobacteriaceae	139	140	Decreased
3	Actinobacteria_Actinobacteria_Bifidobacteriales_Bifidobacteriaceae	145	140	Decreased
4	Bacteroidetes_Bacteroidia_Bacteroidales_p-251-o5	147	140	Increased
5	Bacteroidetes_Bacteroidia_Bacteroidales_Bacteroidales_RF16_group	147	138	Increased
6	Kiritimatiellaeota_Kiritimatiellae_WCHB1-41_uncultured_rumen_bacterium	151	131	Decreased
7	Spirochaetes_Spirochaetia_Spirochaetales_Spirochaetaceae	143	130	Decreased
8	Tenericutes_Mollicutes_Anaeroplasmatales_Anaeroplasmataceae	144	130	Increased
9	Epsilonbacteraeota_Campylobacteria_Campylobacterales_Campylobacteraceae	not significant	129	Decreased
10	Tenericutes_Mollicutes_EMP-G18_uncultured_bacterium	129	129	Increased
11	Verrucomicrobia_Verrucomicrobiae_Verrucomicrobiales_Akkermansiaceae	127	128	Decreased
12	Tenericutes_Mollicutes_Izimaplasmatales_gut_metagenome	141	127	Increased
13	Bacteroidetes_Bacteroidia_Bacteroidales_Bacteroidaceae	141	126	Decreased
14	Bacteroidetes_Bacteroidia_Bacteroidales_Bacteroidales_UCG-001	125	118	Increased
15	Tenericutes_Mollicutes_Izimaplasmatales_uncultured_bacterium	138	117	Decreased
16	Bacteroidetes_Bacteroidia_Bacteroidales_Prevotellaceae	No significant	116	Increased
17	Firmicutes_Clostridia_Clostridiales_Eubacteriaceae	116	113	Decreased

**Table 5 microorganisms-09-02162-t005:** Comparisons of the relative abundances (median, %) of six genera identified by ANCOM analysis that varied significantly between the control and treatment groups by sampling days in the last week (days 21, 22, 24 and 28) of the study period. All are significant (*p* < 0.05).

Family	Genus	Group *	Sampling Days (%)						
			2	7	16	21 *	22	24	28
Enterobacteriaceae	Escherichia-Shigella	Control	7.63	0.00	0.07	0.02	0.05	0.02	0.03
		Trt	0.08	0.03	0.02	0.06	0.00	0.00	0.03
Clostridiaceae_1	Clostridium_sensu_stricto_1	Control	0.03	1.05	1.19	0.24	0.15	0.37	0.14
		Trt	0.14	1.55	0.81	1.34	0.03	0.64	0.63
Acidaminococcaceae	Phascolarctobacterium	Control	0.00	0.08	0.09	0.42	0.16	0.20	0.12
		Trt	0.00	0.13	0.07	0.53	0.12	0.12	0.10
Rhodospirillales	Unassigned	Control	0.00	0.09	0.11	0.02	0.00	0.02	0.04
		Trt	0.00	0.20	0.03	0.03	0.00	0.00	0.00
Ruminococcaceae	Ruminococcaceae_UCG-010	Control	0.00	0.50	0.11	0.57	0.69	0.30	0.17
		Trt	0.00	0.11	0.06	0.41	0.23	0.28	0.16
Erysipelotrichaceae	Turicibacter	Control	0.00	0.15	0.12	0.04	0.01	0.03	0.06
		Trt	0.00	0.37	0.10	0.29	0.00	0.18	0.15

* Enrofloxacin was administered to calves right after fecal collection on Day 21 in treatment groups. Trt—treatment.

**Table 6 microorganisms-09-02162-t006:** Results of ANCOM analysis showing that bacterial genera are significantly different between pre- and post-treatment samples of low and high dosing regimens. High W values indicate significant differences in abundance levels between groups.

Phylum	Class	Genus	Order	W	Pre vs. Post (Change)
**1. Low dose**					
Bacteroidetes	Bacteroidia	uncultured_Bacteroidales_bacterium	Bacteroidales	325	Increased
Bacteroidetes	Bacteroidia	uncultured_bacterium	Bacteroidales	298	Increased
**2. High dose**					
Proteobacteria	Alphaproteobacteria	uncultured_bacterium	Rhodospirillales	344	Decreased
Bacteroidetes	Bacteroidia	Prevotellaceae_UCG-003	Bacteroidales	335	Increased
Firmicutes	Clostridia	Lachnospiraceae_FCS020_group	Clostridiales	316	Increased
Kiritimatiellaeota	Kiritimatiellae	uncultured_rumen_bacterium	WCHB1-41	313	Decreased

**Table 7 microorganisms-09-02162-t007:** Antimicrobial resistance determinants detected in four groups of calves in pre-and post-treatment pooled fecal samples from calves by metagenomic Hi-C.

Antibiotic Class	Control		Low Dose Healthy		High Dose Healthy		High Dose BRD	
	Pre (60 ^a^, 9 ^b^)	Post (43, 5)	Pre (80, 7)	Post (118, 25)	Pre (90, 22)	Post (67, 18)	Pre (39, 5)	Post (123, 22)
Aminoglycoside	aph2(161 ^c^,8 ^d^), aph3(163,9), ant6(132,10), ant9(6,4), sat(72,5)	aph2(47,15), aph3(47,15), ant6(45,8), ant9(40,9), sat(22,8)	aph2(141,23), aph3(153,26), ant6(193,33), ant9(48,13), sat(141,23)	aph2(7,6), aph3(68,8), ant6(3,2), ant9(1,1)	aph2(515,21), aph3(516,22), ant6(65,15), ant9(67,13), sat(115,15)	aph2(99,16), aph3(101,18), ant6(159,14), ant9(150,11), sat(53,11)	aph2(79,11), aph3(79,11), ant6(17,6), ant9(20,8), sat(79,11)	aph2(44,17), aph3(41,15), ant6(42,18), ant9(30,12), sat(8,5)
Beta-lactam	aci(2,1), rob(2,1)	aci(2,1)	aci(17,3), cfx(1,1), rob(12,3)	pbp2(1,1), rob(63,4)	rob(1,1)	aci(1,1), oxa(162,4), rob(2,2)	cfX(2,1)	aci(1,1),cmX(7,2)
Macrolide	ermB(1,1), ermF(1,1), ermG(7,5), ermQ(2,2), mefE(445,4)	ermF(11,1), ermG(2,2), mefE(109,3)	ermB(4,4), ermF(783,12), ermG(36,4), ermQ(1,1), mefE(442,9)	ermF(3,1), ermG(3,3), ermQ(3,2), mefE(86,2)	ermB(2,2), ermG(7,5), ermQ(47,8), ermX(1,1), mefE(11350,51)	ermF(2,2), ermG(63,2), ermQ(1,1), mefE(508,3)	ermG(4,3), ermQ(2,1), mefE(1752,5)	ermB(2,1), ermF(1,1), ermG(3,3), ermQ(87,12), ermX(2,1), mefE(4693,45)
Phenicol	cfr(16,5)	cfr(42,12)	cfr(34,7)	cfr(3,2)	cfr(52,7), floR(1,1)	cfr(13,4)	cfr(9,5)	cfr(29,3)
Sulfonamide	NA ^e^	NA	NA	NA	sulII(1,1)	NA	NA	NA
Tetracycline	tet32(1,1), tet40(118,19), tet44(2,2), tetBP(2,2), tetL(1,1), tetO(14,4), tetQ(457,8), tetW(294,25)	tet32(1,1), tet40(157,18), tetA(5,2), tetB(9,1), tetO(4,3), tetW(248, 23), tetX(1,1)	tet32(2,2), tet40(371,31), tetA(4,3), tetL(1,1), tetM(2,2), tetO(121,23), tetQ(1833,31), tetW(721,56)	tet32(2,1), tet40(32,16), tet44(3,3), tetB(4,1), tetL(1,1), tetM(3,1), tetO(32,15), tetQ(67,4), tetW(186,36), tetX(2,1)	tet32(3,3), tet40(534,46), tet44(69,11), tetA(54,15), tetB(436,21), tetM(23,7), tetO(76,22), tetQ(647,27), tetW(718,49), tetX(1,1)	tet32(21,8), tet40(154,27), tetA(2,2), tetO(46,27), tetW(1371,79)	tet40(48,15), tetA(2,2),tetM(1,1),tetO(67,18), tetQ(27,3), tetW(185,30), tetX(1,1)	tet40(161,40), tet44(6,3), tetA(16,9), tetB(8,4), tetM(11,5), tetO(39,15), tetQ(414,17), tetW(440,63), tetX(1,1)

^a^ Number of total clusters in the pooled sampled, ^b^ number of clusters with ≥80% complete genome, ^c^ number of ARG hits detected in the pooled sample, and ^d^ number of clusters hosting the specified ARG. ^e^ Not available, no ARGs of this antibiotic were detected.

**Table 8 microorganisms-09-02162-t008:** Variations of ARGs between pre- and post-treatment pooled fecal samples in four groups of calves. The control group did not receive any antibiotics.

Group	Both Pre-and Post-Treatment	Only Pre-Treatment	Only Post-Treatment
Control	aph2, aph3, ant6, ant9, aci, ermF, ermG, mefE, cfr, sat, tet32, tet40, tetO, tetW (**total ARGs 14**)	rob, ermB, ermQ, tet44, tetBP, tetL, tetQ (**total 7**)	tetA, tetB, tetX (**total 3**)
Low dose healthy	aph2, aph3, ant6, ant9, ermF, ermG, ermQ, mefE, cfr, rob, tet32, tet40, tetL, tetM, tetO, tetQ, tetW (**total 17**)	aci, cfX, ermB, sat, tetA (**total 5**)	pbp2, tet44, tetB, tetX, emrD (**total 5**)
High dose healthy	aph2, aph3, ant6, ant9, rob, ermG, ermQ, mefE, cfr, sat, tet32, tet40, tetA, tetO, tetW, tetX (**total 16**)	ermB, ermX, sulII, floR, tet44, tetB, tetM, tetQ (**total 8**)	aci, oxa, ermF (**total 3**)
High dose BRD	aph2, aph3, ant6, ant9, ermG, ermQ, mefE, cfr, sat, tet40, tetA, tetM, tetO, tetQ, tetW, tetX (**total 16**)	cfx	aci, cmX, ermB, ermF, ermX, tet44, tetB, tetQ (**total 8**)

**Table 9 microorganisms-09-02162-t009:** Quantitative PCR results for five antimicrobial resistance determinants in three groups of calves, log_2_ transformed mean and standard deviation (SD).

ARGs	Fold Change	Control	Low Dose Healthy	High Dose Healthy
***tet*W**	Mean	−0.21	0.58	0.34
	SD	0.017	0.065	0.015
	*p*-value *	NA	0.022 ^a^	0.359
***tet*O**	Mean	−0.33	0.07	−0.36
	SD	0.040	0.209	0.081
	*p*-value	NA	0.147	0.655
***tet*X**	Mean	−1.30	−2.28	0.23
	SD	0.100	0.095	0.084
	*p*-value	NA	0.180	0.359
***erm*B**	Mean	0.20	−0.43	−1.01
	SD	0.026	0.162	0.018
	*p*-value	NA	0.359	0.022 ^b^
***erm*F**	Mean	−1.06	−1.80	0.28
	SD	0.234	0.055	0.106
		NA	0.180	0.359

* The Dunn test—comparing the treatment groups against the control group with the log_2_ transformed values. ^a^
*tet*W significantly increased in the low dose healthy group post-treatment pooled samples compared to the control group. ^b^
*erm*B significantly decreased in the high dose healthy group post-treatment pooled samples compared to the control group.

## Data Availability

Data are available on request.

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
