# Peer review of "Enrofloxacin Alters Fecal Microbiota and Resistome Irrespective of Its Dose in Calves"

_microorganisms, 2021, doi:10.3390/microorganisms9102162_

Round 1
Reviewer 1 Report
In general, the manuscript is well prepared, the materials and methods are well described, the results are properly presented and fully documented, and the discussion is comprehensive. I have only one comment:
- the authors conducted an invasive experiment, culminating in the euthanasia of animals. Although they cited compliance with the relevant procedures (lines 106-108), the ethics committee approval number for carrying out a specific experiment is usually also given. I suggest that the authors fill in this deficiency.
Line 243 and others - I suggest replacing with the word "infected"
Author Response
Thank you for reviewing our paper and providing important suggestions. Please, kindly find our responses to your suggestions below.
The protocol was approved by the Iowa State University Institutional Animal Care and Use Committee (IACUC 8-12-7432-B). The protocol number is now included in line 107.
The word “challenged” was replaced by “infected” in line 244.
Reviewer 2 Report
Interesting paper on a hot topic. The problem of antimicrobial resistance has been put aside during the pandemics, but is is a silent endemia we must deal with.
The authors should avoid mentioning their own results in the introduction. This section should end with the aim of the study.
Why did the authors choose to administer enrofloxacin only eight days after the Mannheimia haemolytica challenge, when all moderate respiratory signs had already decreased? This may have compromised the evaluation of this challenge, and the absence of differences between the two groups may be irrelevant!
It would be interesting to discuss these results from a One Health perspective. How may this increased resistance to enrofloxacin affect the resistance of human to ciprofloxacin, for instance?
Author Response
Thank you for reviewing our manuscript and providing valuable suggestions. We have kindly provided responses to your comments and question below.
The results mentioned in lines 90-102 have been deleted from the introduction section.
Due to the complexity of multiple calves and precise sampling requirements it was decided before the study that 8 days post inoculation was an acceptable timeline to allow bacteria inoculation to instigate disease but not progress to severe chronic pneumonia. Unfortunately, the inoculation of Mannheimia haemolytica did not result in the severity of respiratory disease as was expected.
The implication of the resistome study has been discussed in lines 774-87 in the light of ‘One Health’ perspective.
“A number of the ARGs observed in this study can confer resistance to medically important antimicrobials of humans and animals. Interestingly, treatment with the enrofloxacin was associated with both increased and decreased ARG hits, as well as both increased and decreased host range observed to be carrying the ARG. As evidenced in Figure 6, these dynamics occur even in control animals not exposed to antibiotic treatment, and in some cases (low dose) antibiotic treatment resulted in a net decreased abundance and host range of ARGs. Importantly, the majority of taxa observed to have ARGs in this study are not direct human or animal pathogens, and hence pose limited direct risk of clinically significant treatment failure in a human or animal. However, as demonstrated by this study, the resistome dynamics both in treated and untreated (control) animals are complex and require additional research to understand the drivers and impacts of ARG transmission. Understanding ARG disseminations and minimizing their detrimental impacts require concerted efforts among animal, human, and environmental health expertise, in short, a “One Health” approach [6-8].”